# Nonlinear Integer Programming for Solving Preventive Maintenance Scheduling Problem for Cogeneration Plants with Production

**Khaled Alhamad [1],\* , Yousuf Alkhezi [2] and M. F. Alhajri [1]**

1   The Public Authority for Applied Education and Training, College of Technological Studies, PAAET,
    Kuwait City 70654, Kuwait
2   The Public Authority for Applied Education and Training, College of Basic Education, PAAET,
    Kuwait City 70654, Kuwait
*   Correspondence: km.alhamad@paaet.edu.kw

**Abstract:** Preventive maintenance (PM) is a maintenance program with activities created at a determined interval or according to certain principles, designed to reduce the likelihood of failure or deterioration of item performance. This aims to improve overall reliability and system availability. In this research, a preventive maintenance schedule (PMS) was designed for electricity and desalination of water in power plants, subject to meeting relevant constraints. The proposed methodology is used to generate a PMS for the boilers, turbines, and distillers. A nonlinear integer programming (NLIP) model was employed to address this problem. The results of the proposed method were compared with the PMS for a power station in Kuwait. The results were better in terms of the volume of production and in terms of the gap between the available production and demand in order to continue providing consumers with electricity and water without a shortage in the event of a breakdown in equipment. It produces an improvement of 12.12% and 16.58% respectively, for water and electricity. Furthermore, the sensitivity and robustness of the proposed method were analysed by increasing the maintenance duration for some equipment, increasing the demand, and adding various additional conditions. In addition, a comparison of additional conditions with a binary problem method in terms of computer time for the search for an optimal solution was carried out, where the model provided an optimal solution in a reasonable time. Among the most important benefits that the user can obtain for this technique are extending the life of the equipment, increasing efficiency, and reducing expenses.

**Keywords:** preventive maintenance; scheduling; optimization; zero–one integer programming; nonlinear programming





## 1. Introduction

Preventive maintenance (PM) is described as frequent maintenance to help keep equipment up and operating. It needs thorough planning and the scheduling of equipment maintenance before an actual problem exists. Preventive maintenance can be very complicated for companies that have a lot of equipment. The benefits of preventive maintenance are many, for example, saving the company from an expensive repair due to an unexpected equipment malfunction that must be repaired quickly, as well as extending the life of the equipment of the company, reducing exposure to human injuries due to breakdowns, improving equipment reliability, and much more. PM continues to be the number one priority for manufacturers. For example, in 2020, 76% of companies in the manufacturing industry worldwide prioritized preventive maintenance, and 60% of the manufacturing industry performs reactive maintenance. For the latest maintenance statistics, see [1].

A preventive maintenance schedule (PMS) is the schedule by which maintenance times for equipment or devices in an organization are determined, within a specific time horizon.

The PMS issue is usually intended to be solved economically or in terms of improving reliability. Economically, this occurs by reducing the total operating expenses during a time horizon for scheduling, while many reliability indicators include expected shortage of net peak reserves, projected energy unavailable, and loss of probability of load. The general problem with the PMS is obtaining the best sequence of maintenance procedures for each component of the system in each period during the planning horizon [2,3].

There are many areas in which preventive maintenance is applied, for example, the transportation field, such as a fleet of trains [4–6], planes [7,8], ships [9–11], and buses [12–14], which need preventive maintenance periodically. In addition, factories [15–19] that contain machines and devices used in industry need preventive maintenance. Moreover, oil companies and power stations apply PM for their equipment [20–24]. The purpose of the PMS function is clearly to contribute to achieving the goals of the organization, which, in turn, requires the need for practical maintenance to be consistent with the organization objective. For more, see [25].

Cogeneration is defined primarily as the production of several forms of energy, such as heat, electricity, and fresh water. Desalination refers to some of the processes used to produce fresh water suitable for human consumption. The process is carried out by removing a certain amount of salt from sea water. Electricity and desalination of water can be generated in several ways—through nuclear, thermal, wind, or hydroelectric energy. All power stations apply PMS to avoid any type of production shortage, which affects supply due to equipment downtime.

Since this type of problem considered is a mixed integer, non-convex, large-scale combinatorial optimization problem, many methods in previous decades were used to address it, such as different types of deterministic [17,26–28], heuristic [29–33], or hybrid methods [3,34]. Heuristic methods are widely used for solving PMS; for instance, Doost-parast et al. [35] applied a simulated annealing algorithm for systems with deteriorating components. The objective was to maintain a certain level of reliability with a minimum total cost of related maintenance. Some illustrative examples were addressed with the aim of evaluating the performance of the proposed approach. Wang and Lin [36] applied a particle swarm optimization (PSO) to minimize a periodic preventive maintenance cost for a series–parallel system. Duarte et al. [26] presented the particle swarm optimization (PSO) algorithms and the genetic algorithms (GA) for a power system using wind generation. The goal was to allow power system operators to have a maintenance schedule that reduces the likelihood of a potential loss of load as much as possible in the power system.

Alhamad et al. [23] applied genetic algorithms for a multi-cogeneration power system to create two preventive maintenance schedules, one for electricity and one for distillers. The aim of this model was to maximize the available number of operational units in each station. The proposed model has been applied to the power station in the State of Kuwait. Furthermore, integer programming is employed for solving the PMS problem. Perez-Canto and Rubio-Romero [37] considered a problem in the wind power plant. The goal was to conduct the PMS from a reliability perspective, as system reliability was maximized. The given linear model contains many variables and constraints. A real case study was presented, which was based on a real energy system in Spain, where the proposed analysis has been validated. Mollahassani-Pour et al. [38] have developed a new formula for a PMS associated with the new cost reduction index (CRI) for generating units. A linear mixed integer program (MILP) was used to minimize operating costs as well as to minimize the maintenance cost over a specific time horizon. Some constraints were relaxed and degraded into several sub-problems. Canto [39] developed the Benders' decomposition technique to solve a model of a real power station. The author proposed two optimization approaches, one to minimize the cost, while the second was to maximize the reliability, meeting a set of constraints. A realistic energy system was applied to verify the efficiency of the proposed analysis. Alhamad and Alhajri [24] addressed the problem of electricity and distiller plants with production using zero–one integer programming. Mollahassani-pour et al. [40] focused on minimizing total operating and maintenance expenses for a power system using mixed integer programming. Ghaffarpour et al. [41] conducted a study of the water and power plant using mixed integer linear programming in order to minimize the

total costs in addition to increasing the reliability subject to the maintenance constraints and the operating area. The effectiveness of the proposed method has been validated by implementation in a real remote area.

According to the literature, most of the research published utilized heuristic, linear, or mixed integer programming approaches to solve these types of problem. On the other hand, there is a lack of research that uses non-linear programming to tackle the PMS problem due to the convexity or non-convexity of the problem, where it is difficult to solve the large-scale problems using available solvers. The following is a display of PMS publications in general and not limited to power plants. There is some research in applied non-linear programming. For example, Gagnon et al. [42] tackled the problem of a hydroelectric system by nonlinear programming (nonconvex), with the goal of minimizing the energy shortages. The model includes storage variables, while water drainage and spillage variables are implicit. Metaheuristics have been explored due to the non-convexity generally associated with generating a power plant function.

Shuya et al. [43] addressed an offshore wind farm, where the objective function was to schedule the preventive maintenance of offshore turbines to satisfy the power supply without failure. A non-linear multi-objective programming model was proposed, where two significant goals were considered. One was to maximize the system reliability, while the other was to minimize the maintenance related cost. Guedes et al. [44] proposed a new non-linear model for cascading hydroelectric generation and preventive maintenance scheduling. In order to reduce the complementary thermal generation and increase the value of water in the future to the maximum, the researchers improved the level of the tank and the scheduling of maintenance simultaneously. The model was applied in Brazilian waterfalls. Rezaei-Malek et al. [45] developed a correct mixed non-linear programming model for integrated planning to check the quality of parts and preventive maintenance activities. That study considered the multi-stage chain manufacturing system while the non-linear production stages deteriorated. Deterioration assumes a non-linear process, and an approximate approach on the basis of detachable programming is developed to deal with the nonlinearity. The proposed model provides optimal times and locations for preventive maintenance while reducing the overall manufacturing cost. Wu and Zuo [46] presented a review of models of preventive maintenance and investigation of their interrelations, where the researchers classified these models into three categories: linear, nonlinear, and a hybrid of both. The potential extensions of these three PM models were also discussed.

This research study will focus on one power station in Kuwait, where the study provides a PMS for each piece of equipment located in the power plant. The methodology is a non-linear integer programming problem. The method designed to produce an optimal maintenance schedule for cogeneration plants in terms of maximizing the available pieces of equipment in each unit for a 12-month demand cycle. The main objective of this paper is to set a schedule for the maintenance equipment in a way that maximizes utilization of the equipment while minimizing power outages. Since maintenance scheduling is usually an optimization problem, this problem is subject to a number of constraints, such as maintenance window limitations, time limits, and output load limits for water and power. The main objective of this optimization problem is to allow power generation and water production to meet demand without any interruption by maintaining the required equipment available during operation.

The paper is organized as follows: In Section 2, the problem background is provided. In addition, the section introduces a description of the problem and reviews the formulation used to solve the problem, where a nonlinear integer programming (NLIP) model with constraints for the PMS of power station is formulated. In Section 3, computational tests are presented, where Test 1 examines the validation of the proposed approach and compares it with the schedule of the Ministry of Electricity and Water in Kuwait. Test 2 is a sensitivity analysis to examine the efficiency and robustness of the proposed approach. Finally, the conclusions in addition to further work to solve this type of problem are presented in Section 4.

There is great confidence that the model proposed in this paper will be reliable and able to create an ideal schedule for the industrial sectors.

## 2. Problem Description

### 2.1. Problem Background

The main source of energy that we get from the electricity (and fresh water) consumed in Kuwait is still the chemical energy found in the fuel that is made up of gas and liquid oil products. The process of transforming the primary energy of fuel into electrical energy passes through several stages within the power plants (and desalination plants), which includes complex equipment and plants requiring huge financial investments. These include very large boilers that burn vast amounts of fuel and transform the chemical energy into heat energy, which, in turn, produces large amounts of high-pressure hot steam. This steam drives steam turbines that transform thermal energy into chemical energy. Spin generators that transform mechanical energy into electrical energy are used to export it to the network for transmission, distribution, and delivery to consumers.

The Ministry of Electricity and Water in Kuwait is responsible for supplying power and water. In Kuwait, plants consist of a boiler, a turbo-generator, and a distiller. The boiler produces high pressure steam from fuel, the turbo-generator produces power from high pressure steam, and the distillers produce desalinated water from salt water using low pressure steam. This equipment requires regular maintenance to maintain their continuity at work and to avoid any breakdown, which may cause a lack of electricity or water, in addition to the high cost of repair. On the other hand, maintenance should be scheduled in a correct and scientific manner so that maintenance is not concentrated in peak periods of demand. Obviously, preventive maintenance involves a basic trade-off between high-peak and off-peak.

### 2.2. Mathematical Formulation

The problem of PMS addressed in this paper is formally defined as follows: first, since there is only one plant, a set of plants will not be mentioned, and we move to the set of units. Let $U$ be a set of units with $u = \{1,...,U\}$, where $U$ is the total number of units. Each unit contains $n$ number of different types of equipment, where there are three types of equipment in each unit: boiler, distiller, and turbine. Each piece of equipment must undergo maintenance for a period of time, where more than one piece of equipment can undergo maintenance at a time. When the boiler equipment is in maintenance, the turbine equipment and the distiller equipment are suspended until the boiler equipment maintenance process is completed. On the other hand, the maintenance of any turbine or distiller equipment does not require stopping the boiler equipment unless the maintenance of the turbine equipment and the distiller equipment are performed simultaneously. Maintenance periods for equipment are not necessarily identical. Throughout maintenance, a minimum number of pieces of equipment in the plant must remain operational. Meanwhile, increasing the gap between available production and demand during the time horizon plays a major role in avoiding a shortage of production, whether it be electricity or water, due to an emergency breakdown of a number of pieces of equipment.

The objective function of this problem is to schedule the preventive maintenance tasks for all equipment in a way that (1) increases the number of pieces of equipment available over the operational planning time horizon, $T$, while, at the same time, minimizing power outages which affects the fulfillment of consumer demands, and (2) can be achieved by closing the gaps between demand and production for the entire time horizon. Meanwhile, a number of constraints need to be considered, which are described in Section 3.3.

### 2.3. Mathematical Formulation

The nomenclature for NLIP approach is divided into sets, data, and decision variables, as follows:

Sets:

$u = 1..U$ Denotes units

$b = 1..B$ Denotes boilers

$r = 1..R$ Denotes turbines

$d = 1..D$ Denotes distillers

$t = 1..T$ Denotes time periods

Data:

$\overline{D}_{ub}$ Duration of maintenance for boiler $b$

$\overline{D}'_{ur}$ Duration of maintenance for turbine $r$

$\overline{D}''_{ud}$ Duration of maintenance for distiller $d$

$ET_{ub}$ Earliest start time for boiler $b$

$ET'_{ur}$ Earliest start time for turbine $r$

$ET''_{ud}$ Earliest start time for distiller $d$

$LT_{ub}$ Latest start time for boiler $b$

$LT'_{ur}$ Latest start time for turbine $r$

$LT''_{ud}$ Latest start time for distiller $d$

$PR_{ur}$ Maximum production capacity for turbine $r$

$PR'_{ud}$ Maximum production capacity for distiller $d$

$OP_b$ Maximum number of boilers $b$ allowed to be in maintenance

$OP'_r$ Maximum number of turbines $r$ allowed to be in maintenance

$OP''_d$ Maximum number of distillers $d$ allowed to be in maintenance

$DE_t$ Electricity demand

$DW_t$ Water demand

$WRS$ Initial reservoir level of water

$WMIN$ Minimum reservoir level of water

$WMAX$ Maximum reservoir level of water

$THR$ Available human resources

$MAN_{ub}$ Human resources required for maintenance for boiler $b$

$MAN'_{ur}$ Human resources required for maintenance for turbine $r$

$MAN''_{ud}$ Human resources required for maintenance for distiller $d$

$\mu$ A balancing factor between the production of electricity and water, used in the main objective function

$\gamma$ A balancing factor for the gap $ge_t$, used in the main objective function

$\delta$ A balancing factor for the gap $gw_t$, used in the main objective function

Decision Variables:

$$x_{ubt} = \begin{cases} 1, & \textit{boiler b is in maintenance.} \\ 0, & \text{otherwise} \end{cases}$$

$$x'_{urt} = \begin{cases} 1, & \textit{turbine r is in maintenance.} \\ 0, & \text{otherwise} \end{cases}$$

$$x''_{udt} = \begin{cases} 1, & \textit{distiller d is in maintenance.} \\ 0, & \text{otherwise} \end{cases}$$

$$xs_{ubt} = \begin{cases} 1, & \textit{boiler b first week for maintenance.} \\ 0, & \text{otherwise} \end{cases}$$

$$xs'_{urt} = \begin{cases} 1, & \textit{turbine r first week for maintenance.} \\ 0, & \text{otherwise} \end{cases}$$

$$xs''_{udt} = \begin{cases} 1, & \textit{distiller d first week for maintenance.} \\ 0, & \text{otherwise} \end{cases}$$

$$y_{urt} = \begin{cases} 1, & \textit{turbine r is in maintenance or idle.} \\ 0, & \text{otherwise} \end{cases}$$

$$y'_{udt} = \begin{cases} 1, & \textit{distiller d is in maintenance or idle.} \\ 0, & \text{otherwise} \end{cases}$$

$WMAX \geq res_t \geq WMIN$   Available water reserves.$ave_t \geq DE_t.avw_t \geq DW_t.$

Electricity gap between the demand of electricity and available *MW* (Mega Watt) in *kth* week $t$. $ge_t \geq 0$.

Water gap between the demand of water and available *MIGD* (Million Imperial Gallons per Day) in *kth* week $t$. $gw_t \geq 0$.

### 2.4. Mathematical Model

The problem can then be formulated as the following: objective function

*max z:*

$$\sum_t (ave_t + \mu.avw_t) - \sum_t (ge_t/\gamma)^2 - \sum_t (gw_t/\delta)^2 \tag{1}$$

*s.t:*

$$\sum_{t=1}^{T-\overline{D}_{ub}+1} xs_{ubt} = 1 \quad \forall u, b \tag{2}$$

$$\sum_{t=1}^{T-\overline{D}'_{ur}+1} xs'_{urt} = 1 \quad \forall u, r \tag{3}$$

$$\sum_{t=1}^{T-\overline{D}''_{ud}+1} xs''_{udt} = 1 \quad \forall u, d \tag{4}$$

$$\sum_{t'=t}^{t+\overline{D}_{ub}-1} x_{ubt} \geq \overline{D}_{ub}\, xs_{ubt} \quad \forall u, b, t = 1..T - \overline{D}_{ub} - 1 \tag{5}$$

$$\sum_{t'=t}^{t+\overline{D}'_{ur}-1} x'_{urt} \geq \overline{D}'_{ur}\, xs'_{urt} \quad \forall u, r, t = 1..T - \overline{D}'_{ur} - 1 \tag{6}$$

$$\sum_{t'=t}^{t+\overline{D}''_{ud}-1} x''_{udt} \geq \overline{D}''_{ud}\, xs''_{udt} \quad \forall u, d, t = 1..T - \overline{D}''_{ud} - 1 \tag{7}$$

$$\sum_t x_{ubt} \leq \overline{D}_{ub} \quad \forall u, b \tag{8}$$

$$\sum_t x'_{urt} \leq \overline{D}'_{ur} \quad \forall u, r \tag{9}$$

$$\sum_t x''_{udt} \leq \overline{D}''_{ud} \quad \forall u, d \tag{10}$$

$$\sum_u \sum_b x_{ubt} \leq OP_b \quad \forall t \tag{11}$$

$$\sum_u \sum_r x'_{urt} \leq OP\prime_r \quad \forall t \tag{12}$$

$$\sum_u \sum_d x''_{udt} \leq OP''_d \quad \forall t \tag{13}$$

$$y_{urt} \geq x_{ubt} \quad \forall u,b,r,t \tag{14}$$

$$y_{urt} \geq x'_{urt} \quad \forall u, r, t \tag{15}$$

$$y_{udt0} \geq x_{ubt} \quad \forall u,b,d,t \tag{16}$$

$$y'_{ubt} \geq x''_{ubt} \quad \forall u, b, t \tag{17}$$

$$ET_{ub} \leq t.xs_{ubt} \leq LT_{ub} \quad \forall u,b,t \tag{18}$$

$$ET'_{ur} \leq t.xs'_{urt} \leq LT'_{ur} \quad \forall u, r, t \tag{19}$$

$$ET''_{ud} \leq t.xs''_{udt} \leq LT''_{ud} \quad \forall u, d, t \tag{20}$$

$$ave_t = \sum_r \sum_u PR_{ur}(1 - y_{urt}) \quad \forall t \tag{21}$$

$$avw_t = \sum_d \sum_u PR'_{ud}(1 - y\prime_{udt}) \quad \forall \tag{22}$$

$$ge_t = ave_t - DE_t \ \forall t \tag{23}$$

$$gw_t = avw_t - DW_t \ \forall t \tag{24}$$

$$res_0 = WRS \tag{25}$$

$$res_t = avw_t + res_{t-1} - DW_t \ \forall t \tag{26}$$

$$\sum_u \sum_b MAN_{ub} \cdot x_{ubt} + \sum_u \sum_r MAN'_{ur} \cdot x'_{urt} + \sum_u \sum_d MAN''_{ud} \cdot x''_{udt} \leq THR \quad \forall t \tag{27}$$

The objective function (1) maximizes the amount of production of electricity and water, and the production of water has been amplified by the value of $\mu$ in order to approach the production of electricity. Then, the search for the maximum value is equal for electricity and water, and there is no bias for electricity without water. Likewise, $\gamma$ and $\delta$ were used to provide balance during the search to prevent biasing for gaps or for production. In addition, the total gap has been reduced in all periods, making the gaps close together, and there would be no very small gaps in some periods during the time horizon, where the chance of breakdowns is large. Constraints (2)–(4) ensure that each equipment has undergone one maintenance during the time horizon, where Constraint (2) is related to boilers, Constraint (3) is related to turbines, and Constraint (4) is related to distillers. Constraint (5) ensures that maintenance times for boilers are in sequential weeks and are not separate, likewise for Constraint (6) for turbines and Constraint (7) for distillers. Equation (8) ensures that each boiler adheres to the maintenance duration, similarly for Constraint (9) for turbines and Constraint (10) for distillers. To confirm compliance with the upper limit on the number of boilers allowed for maintenance during the same period, Constraint (11) has been stated, similarly for turbines in Constraint (12) and Constraint (13) for distillers. Constraints (14) and (15) require that the distiller be in a state of stoppage, either for maintenance or idle due to boiler maintenance, and the same is true in Constraints (16) and (17) for turbines.

Equation (18) ensure that every boiler maintenance must occur in the specified time window for starting maintenance, likewise for Equation (19) for turbines and Equation (20) for distillers. Constraint (21) calculates the available production of electricity, and Constraint (22) calculates the available production of water. Constraint (23) calculates the gap between available production and the demand for electricity, while Constraint (24) calculates the gap between available production and the demand for water. Furthermore, Constraint (25) assumes that the available water before starting the implementation of the PMS is equal to the initial reservoir, while Constraint (26) calculates the reservoir water during the time horizon. Finally, Constraint (27) is set to ensure that the labor required for maintenance does not exceed the available human resources.

## 3. Computational Tests

### 3.1. General Data

The input data is provided by the Ministry of Electricity and Water in Kuwait State. The computational test is based on 1 plant and 8 units, where each unit consists of 4 pieces of equipment, 1 boiler, 2 distillers, and 1 turbine, which means there are 32 pieces of equipment for the entire plant. The aim of this research is to reduce the power outages or water shortages to a large extent in the event of one or more equipment failures during the production process. Therefore, the objective of this research was to increase electricity and water production, which means maximizing the availability of operational equipment per unit during the operational planning horizon as well as maximizing the minimum gap across the time horizon between demand and available production.

The model is executed using the CPLEX 12.9.0 optimization solver (IBM Academic Initiative) using a *DELL* desktop computer, core$^{TM}$ i7-1065G7 CPU @ 1.50 GHz, 16.0 GB

RAM. The representative time horizon specified is 1 year. This time horizon can be set to 12 months or 52 weeks, as the month consists of 4 or 5 weeks.

The NLIP problem has been formulated to generate an optimal PMS for a power station in Kuwait state. A set of data that will be considered in this problem is presented as follows:

i. The preventive maintenance window is open for all equipment; therefore, the time window for boilers starts from the first week ($ET_{ub} = 1$) and continues to the last week 52 ($LT_{ub} = 52$). The same applies for distillers and turbines; thus, preventive maintenance work can be carried out for all equipment during the time period: [1,52].

ii. The number of units are eight units ($U = 8$), while the number of pieces of equipment in each unit is four ($n = 4$), one boiler, two distillers, and one turbine. This means that there are 32 equipment involved in this problem.

iii. Maintenance of more than two pieces equipment of the same type (boiler, distiller, or turbine) cannot be performed at the same time.

iv. There is no limit to the manpower, so human resources are available throughout the operational planning period.

Figure 1 displays the volume of demands for electricity and water within one year for the Al-Zour station in the State of Kuwait. Table 1 illustrates the production capacity for each piece of equipment in the station, where D1 and D2 represent distillers No. 1 and 2, while T represents the turbine for each of the eight units in the station.

Two tests were applied to the proposed model, the first test to measure the efficiency and the validation of the proposed model, and this will be detailed in Section 3.2. In Section 3.3, an assessment of the sensitivity and durability of the proposal is performed through increasing the maintenance duration for some equipment and the volume of demand, in addition to some further conditions.

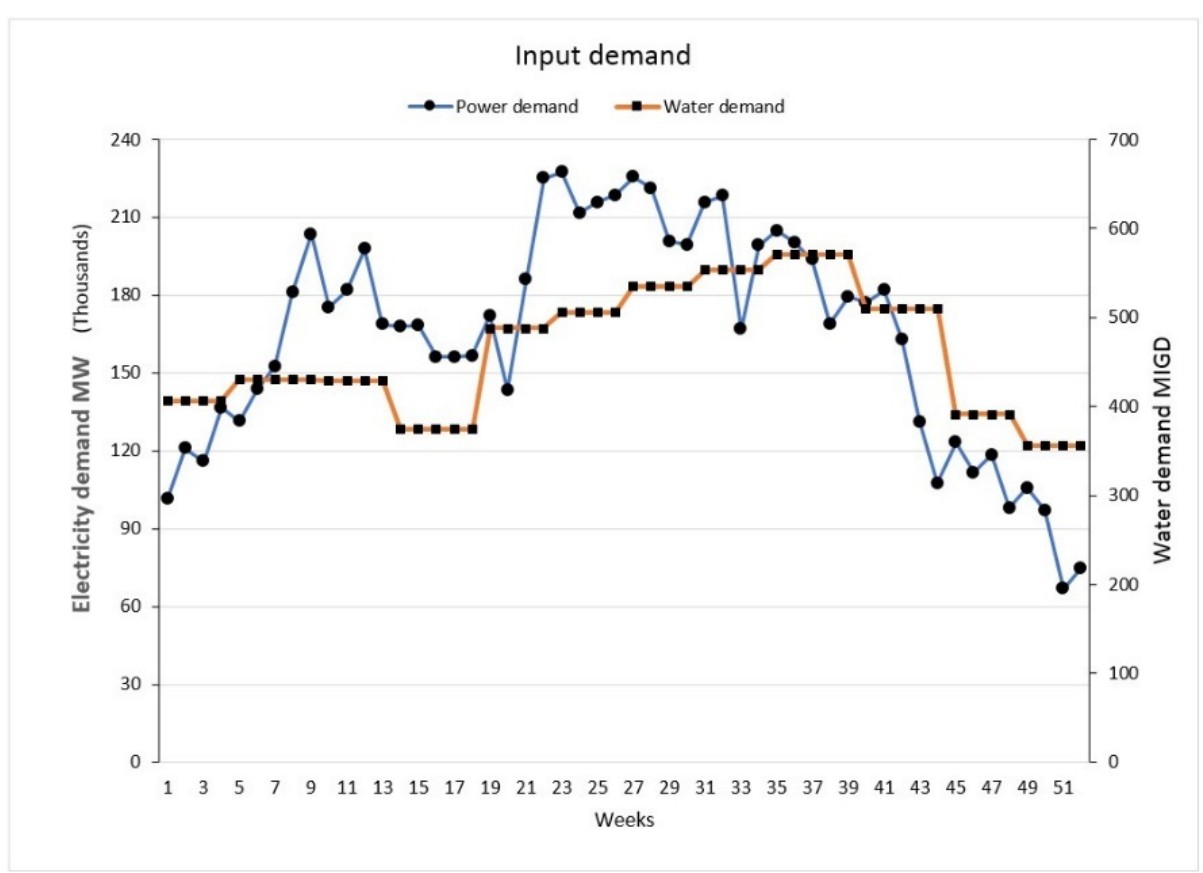

**Figure 1.** Input demand for the model.

**Table 1.** Input data.

| Unit | Equipment | Production | Unit | Equipment | Production |
|------|-----------|-----------|------|-----------|-----------|
|   | D1 | 50.4 | 5 | D1 | 50.4 |
| 1 | D2 | 50.4 |   | D2 | 50.4 |
|   | T | 47,040 |   | T | 47,040 |
|   | D1 | 50.4 | 6 | D1 | 50.4 |
| 2 | D2 | 50.4 |   | D2 | 50.4 |
|   | T | 47,040 |   | T | 47,040 |
|   | D1 | 50.4 | 7 | D1 | 42 |
| 3 | D2 | 50.4 |   | D2 | 42 |
|   | T | 47,040 |   | T | 47,040 |
|   | D1 | 50.4 | 8 | D1 | 42 |
| 4 | D2 | 50.4 |   | D2 | 42 |
|   | T | 47,040 |   | T | 47,040 |

MIGD for distiller; MW for turbine.

### 3.2. Model Validation and Analysis: Test 1

The aim of this study is to obtain reliability preventive maintenance schedules through solution quality and within a reasonable CPU time. The objective function of this research is to maximize electricity and water production, at the same time making the gap between demand and available production at all periods of the time horizon very close, while all constraints are met. Table 2 illustrates the parameters related to the proposed model, such as the number of variables, the optimal solution, the time taken to reach the solution, etc. Table 3 displays all the results for the PMS for the proposed model and compares it with the PMS for the Ministry of Electricity and Water for the State of Kuwait. First, this result has been achieved in a reasonable time of 435 s. The objective function is equal to 36,031,316, and this number is a combination of a set of variables. There are two significant parts of the objective function, one related to the production for both electricity and water (*ave* and *avw*) and the other related to the gap (*ge* and *gw*). Regarding the first part, it consists of available production for both electricity and water (*ave* and *avw*).

**Table 2.** Proposal model result analysis for Test 1.

| Schedule | Var. | Binary | Row | Time * | Obj. Funt. | $\mu$ | $\delta$ | $\gamma$ |
|----------|------|--------|-----|--------|-----------|-------|----------|----------|
| Proposed model | 4316 | 4160 | 6408 | 423 | 36,031,316 | 435 | 9 | 1000 |

* Time in seconds.

Table 3 shows that the volume of electricity production is equal to 17,687,040 MW and that the volume of water production is equal to 36,321.6 MIGD. This available production for both is optimal; this is because the period of preventive maintenance of the turbines and distiller equipment have coincided with the period of preventive maintenance of the boilers. Meanwhile, the duration of preventive maintenance for the turbines is less than the maintenance duration of boilers and distillers by one week, as boiler and distiller maintenance requires 5 weeks, while turbine maintenance requires only 4 weeks; this is the reason that all turbines are idle for a week, as shown in Figure 2 and Figure 5. In contrast, there is no distiller equipment in an idle state throughout the time horizon, since the time period for both boiler and distiller is equal, as shown in Figures 3 and 4.

The value of $\mu$ is 435, as illustrated in Table 2; this value equals the ratio of total electricity demand to total water demand. This factor was added to provide a balance during the search for the maximum value of production on an equal basis to that of electricity and water, included to prevent the search from being more biased for electricity. On the other hand, the total production for PMS for the Ministry of Electricity and Water is equal to 17,498,880 MW, which is lower than the total production for the proposal model schedule, as illustrated in Table 3. This is a result of one of the turbines in the schedule of the Ministry (Turbine 7, as shown in Figure 3 and Figure 7) being in an idle state due to

the inconsistency of its maintenance period with Boiler 7. Consequently, the total turbine shutdown in the proposed model schedule is 40 weeks, of which 32 weeks are for PM, and 8 weeks are idle weeks due to the boiler maintenance period. On the other hand, in the schedule of the Ministry of Electricity and Water, the number of weeks that the turbines are shut down is 44 weeks, 32 weeks for preventive maintenance and 12 weeks idle due to the maintenance period of the boilers. In contrast, the total amount of water production for the proposed model is equivalent to the production of the Ministry's schedule of 36,321.6 MIGD, as shown in Table 3. The reason is due to the compatibility of maintenance of distillers with the maintenance of boilers, which is 40 weeks, and the absence of any distiller in an idle state.

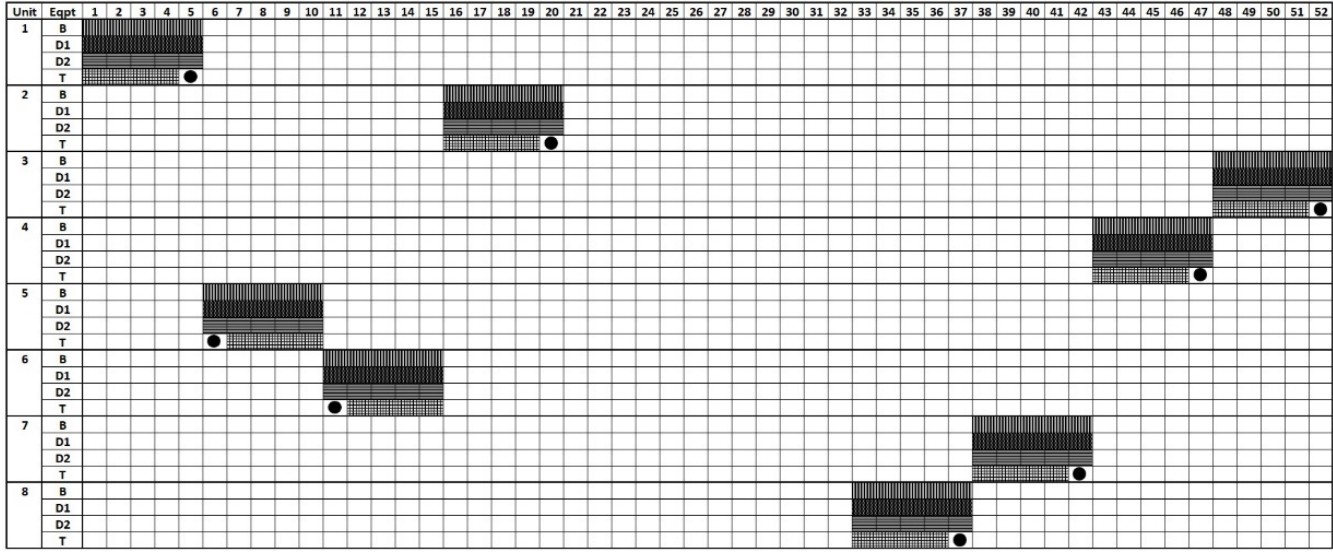

**Figure 2.** PMS for the proposed model. "●" means idle equipment. Top for Boiler, next to down Distiller1, next to down distiller 2, and last one in the bottom refers to turbine.

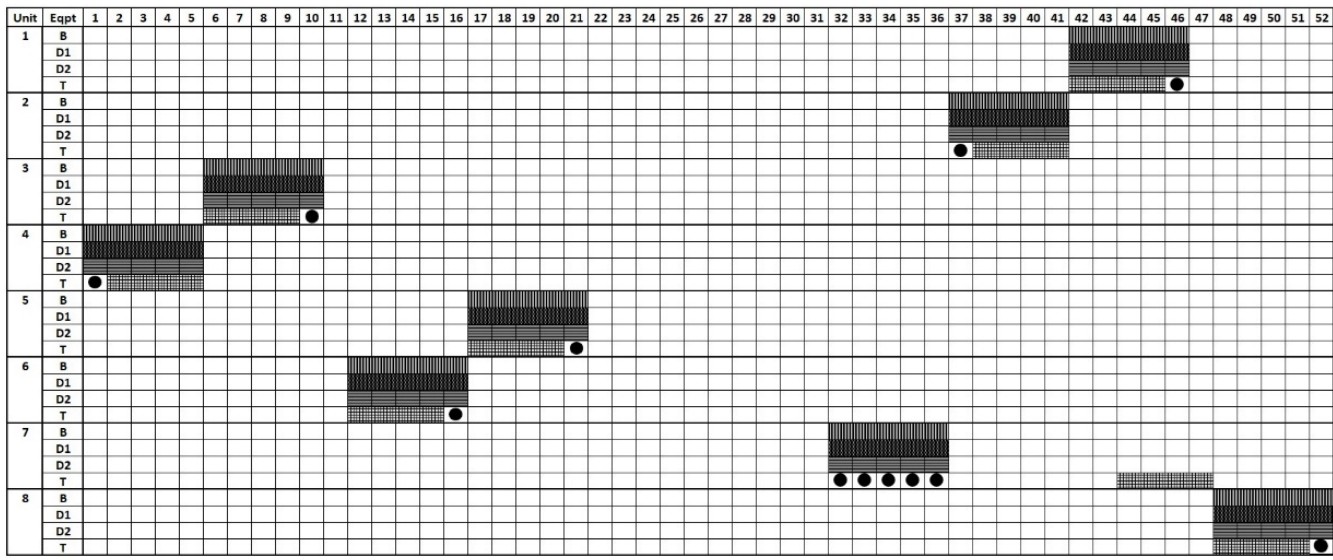

**Figure 3.** PMS for the Ministry of Electricity and Water for the State of Kuwait. Top for Boiler, next to down Distiller1, next to down Distiller 2, and last one in the bottom refers to Turbine.

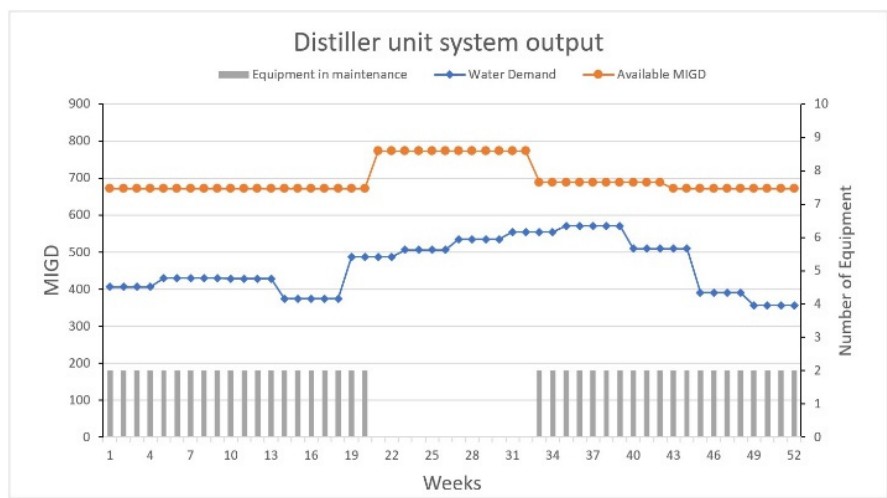

**Figure 4.** Turbine equipment system output.

Regarding the second part of the objective function, Tables 2 and 3 show the gap between available production and demand for both schedules. First, the PMS for the proposal model has two factors, $\gamma$ and $\delta$; the value of $\gamma$ is equal to 1000, while the value of $\delta$ is equal to 9. These values were reached after several experiments to strike a balance during the process of searching for the optimal solution and the non-alignment of methods without the other side. In relation to the lowest gap for electricity, the *ge* value, it is equal to 124,426 MW, while the lowest gap for water, the *gw* value, is equal to 118.1 MIGD, as illustrated in Table 3. On the other hand, the lowest gap values for electricity and water are equal to 110,971 MW and 101.3 MIGD, respectively, for the PMS of the Ministry of Electricity and Water. This means that for electricity availability, the PMS for the proposal model has a larger safety buffer than the PMS for the Ministry of Electricity and Water by 12.12%. As for water availability, the PMS for the proposal model has a larger safety buffer than the PMS for the Ministry of Electricity and Water by 16.58%. It can also be noted that the highest periods of electricity demand are concentrated in the period from week 22 to week 37, and for water, the highest periods are concentrated between week 27 and week 39. This means that the highest period of electricity and water demand is between week 22 and 39. Therefore, it can be noticed that in the PMS for the proposal, there is no maintenance between the 21st week and the 32nd week, which is approximately in the high demand period for electricity and water, as shown in Figures 2, 4 and 5. On the other hand, it can be observed in the PMS of the Ministry of Electricity and Water that there are some weeks when all equipment is in operational condition in low demand times, such as week 11 for electricity and weeks 11 and 47 for water, as shown in Figures 3, 6 and 7.

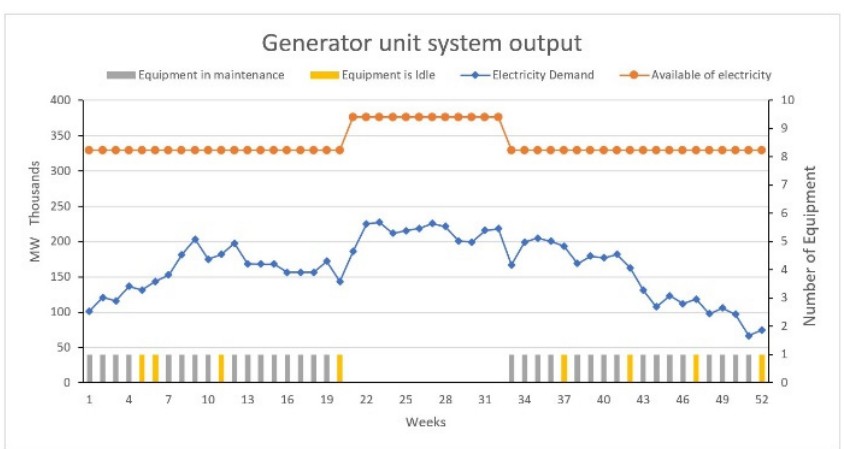

**Figure 5.** Distiller equipment system output.

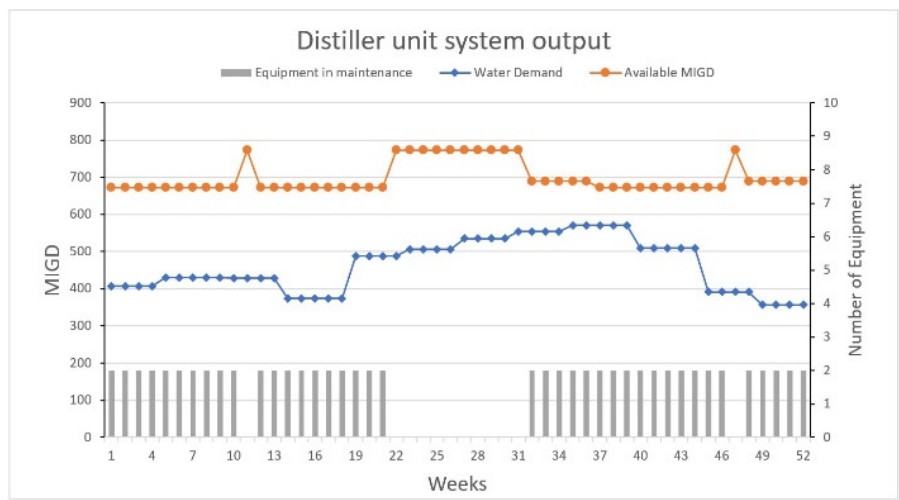

**Figure 6.** Distiller equipment system output for Ministry of Electricity and Water for the State of Kuwait.

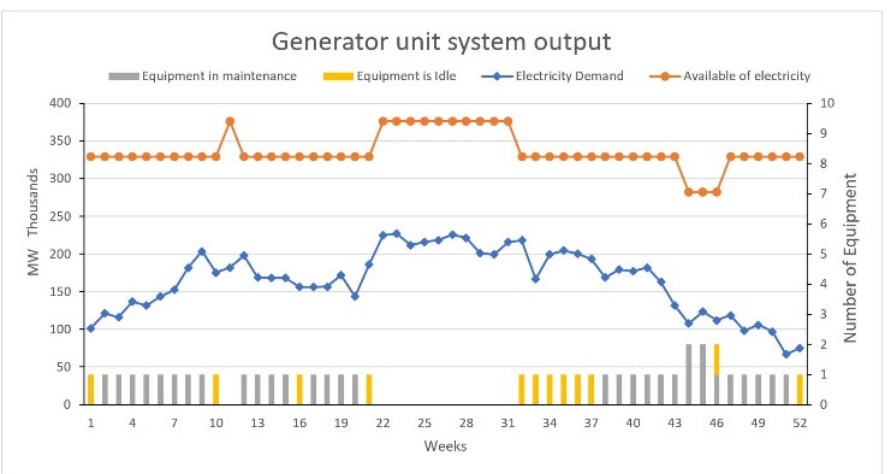

**Figure 7.** Turbine equipment system output for Ministry of Electricity and Water for the State of Kuwait.

**Table 3.** The results of PMS for the proposed model and Ministry of Electricity and Water for the State of Kuwait.

| Schedule | Total Production | | The Lowest Gap | |
|---|---|---|---|---|
| | Electricity (MW) | Water (MIGD) | Electricity (MW) | Water (MIGD) |
| The proposed model | 17,687,040 | 36,321.6 | 124,426 | 118.1 |
| Ministry of Electricity and Water | 17,498,880 | 36,321.6 | 110,971 | 101.3 |

### 3.3. Model Sensitivity and Robustness: Test 2

In this section, a sensitivity analysis was performed to examine the efficiency and robustness of the proposed approach to PMS. First, in order to measure the plants capability to continue production in case of increased demand, we increased production for both the distillers and the turbines, as shown in Figures 8 and 9. It is clear from the figures that the increase available for the distillers is a maximum of 120% and is 150% for the turbines, since an increase of more than this percentage will produce an unfeasible solution. Secondly, we made a comparison between the proposed method in this research and the binary problem (BP) method of article [12] regarding the time taken to reach the optimal solution. According to Figure 10, the average time taken to reach the optimal solution for the NLIP method of the current research is 323 s, with a standard deviation equal to

131 s, where the minimum time is 129 s and the maximum time is 886 s. On the other hand, the average time taken to reach the optimal solution for the BP method is 210 s, with a standard deviation equal to 85 s, where the minimum time is 24 s, and the maximum time is 461 s. It is clear from this experiment that the NLIP method takes a little longer than the BP method, and this is mathematically correct. Conversely, NLIP reaches an optimal solution in a reasonable time, in comparison with the BP method.

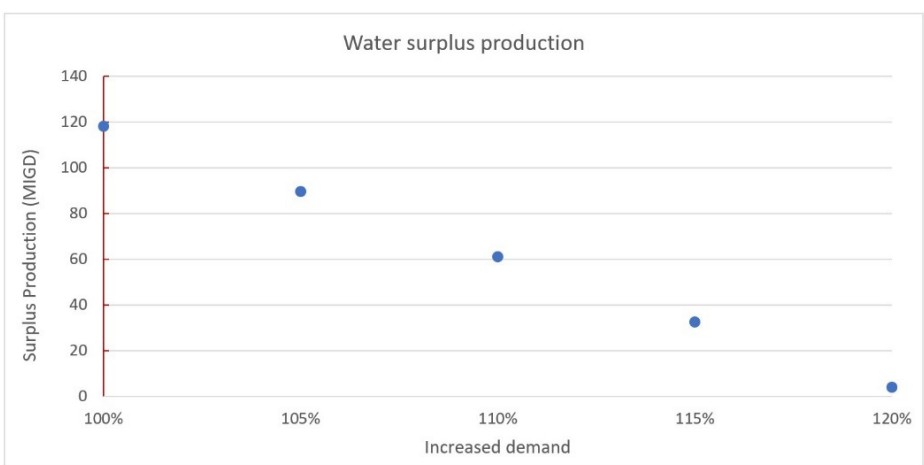

**Figure 8.** Surplus of water production versus increased demand.

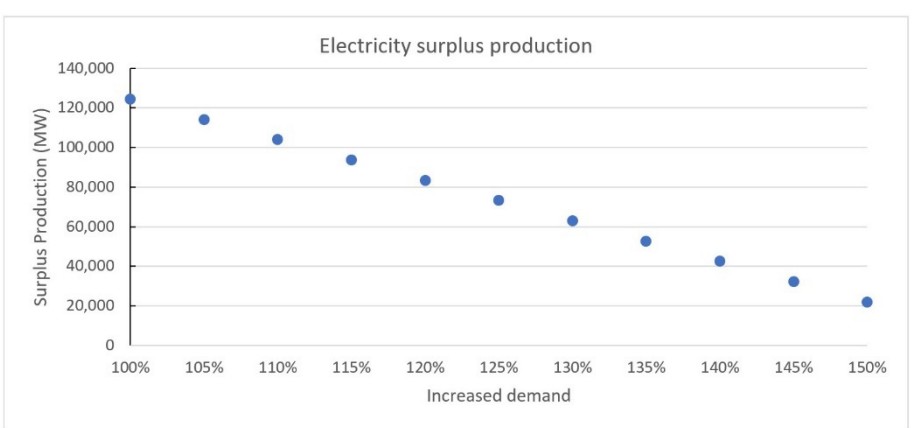

**Figure 9.** Surplus of electricity production versus increased demand.

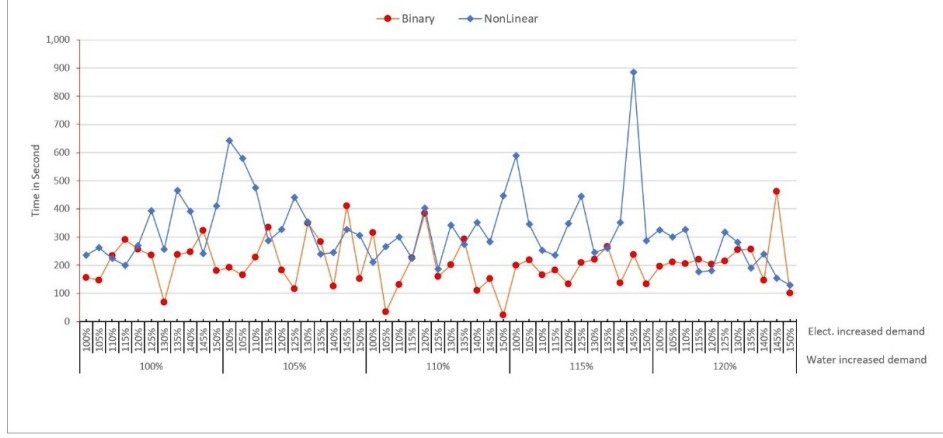

**Figure 10.** Searching time comparison for optimal solution between NLIP and BP.

Moreover, the proposed model contains two main parameters: the demand and maintenance duration, where a scenario is changed with data for each parameter. The volume of water demand is increased by 20%, and the demand for electricity is increased by 40%. On the other hand, assuming that some equipment has a long service life, the duration of preventive maintenance has been increased. Thus, the maintenance duration for some equipment has been increased as follows: Boiler 1 in Unit 6, (D61), 6 weeks; Distiller 1 in Unit 8, $(\overline{D}'_{81})$, 6 weeks; Distiller 2 in Unit 4, $\overline{D}'_{42}$, 6 weeks; Turbine 1 in Unit 2, $\overline{D}''_{21}$, 5 weeks; and Turbine 1 in Unit 7, $\overline{D}''_{71}$, 5 weeks.

In addition, some extra conditions were added, as follows:

i. The time for preventive maintenance of Boiler 1 in Unit 2 must begin before week 17.
ii. The preventive maintenance of Distiller 2 in Unit 6 begins between week 15 and week 47.
iii. The preventive maintenance of Turbine 1 in Unit 8 should start before week 38.
iv. Finally, preventive maintenance of Boiler 1 in Unit 1 cannot begin before Boiler 1 of Unit 5.

In Table 4, the values of $\mu$, $\gamma$, and $\delta$ are 485, 1500, and 9, respectively. The available water production equals 36,128.4 MIGD, and the available electricity production is 17,640,000 MW, as shown in In Table 5. These available productions are optimal for both water and electricity because all maintenance periods were consistent with the boiler maintenance period. This led to the absence of any idle states for all equipment except for Distillers 1 and 2 in Unit 6, as the maintenance duration is one week less than the maintenance duration of the boiler in the same unit. In addition, for most turbines, the maintenance duration is less than the maintenance duration of the boilers by one or two weeks, as shown in Figure 11. It is compulsory, as mentioned previously, that the distillers and turbines must stop operating until the preventive maintenance process of the boiler in the same unit is completed. On the other hand, the production of Test 2 is less than the production of Test 1 for both electricity and water. This is logical because of the extension of the maintenance duration of some equipment, which reflects negatively on the total production.

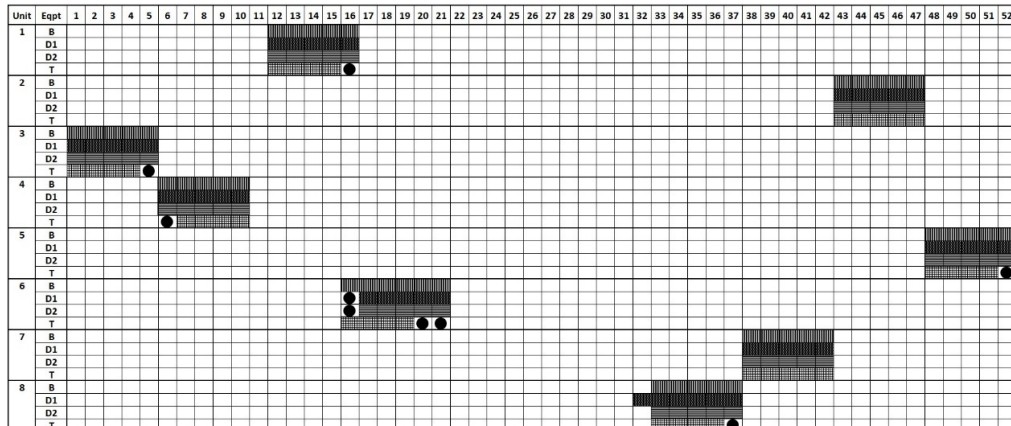

**Figure 11.** PMS for the Ministry of Electricity and Water for the State of Kuwait. Top for Boiler, next to down Distiller1, next to down distiller 2, and last one in the bottom refers to turbine.

In Table 5 and Figure 11, the distribution of preventive maintenance for all distiller equipment and turbines is committed to the period of preventive maintenance that has been identified, in addition to a 20% increase in demand for water and 50% for electricity. It is also obvious that the distribution of maintenance time is concentrated in periods when demand is low and away from peak times of demand. Thus, for distillers, as shown in Figure 12, it can be seen that the equipment operates between week 22 and week 31. However, because the required maintenance schedule is not limited to distillers only but

also accompanies the preventive maintenance of turbines, it can be observed that during the period between week 36 and week 39, the gap has become very narrow, namely 3.96 MIGD. This situation creates a risk in not being able to meet the demand for water if some distillers encounter technical errors during this period. The same applies to turbines, where the equipment is in working mode between week 22 and week 31, but the gap is narrow in week 9 and week 35, namely 42,484.4 MW. Since this is less than the output of one turbine, which is 47,040 MW, as shown in Figure 13, this also creates a risk in failing to meet the demand for electricity if a turbine should encounter technical errors during this period.

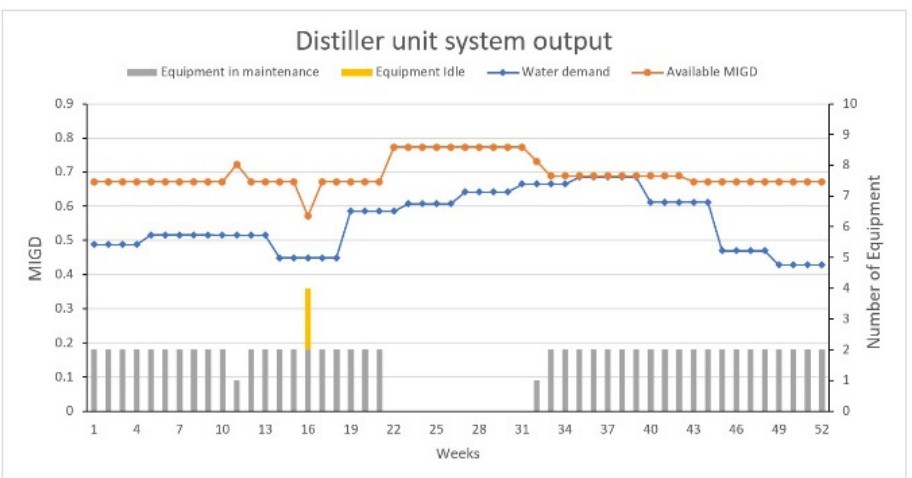

**Figure 12.** Distiller equipment system output.

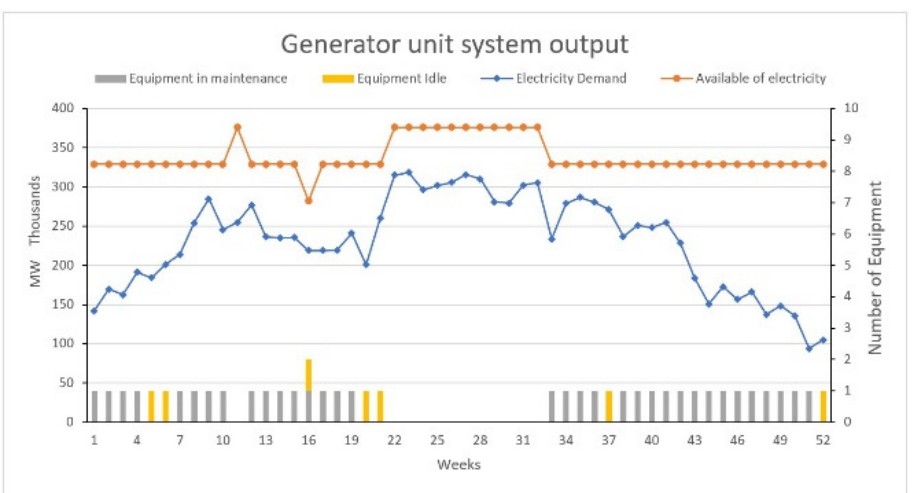

**Figure 13.** Turbine equipment system output.

**Table 4.** Proposal model result analysis for Test 2.

| Schedule | Var. | Binary | Row | Time * | Obj. Funt. | $\mu$ | $\gamma$ | $\delta$ |
|---|---|---|---|---|---|---|---|---|
| Proposed model | 4316 | 4160 | 6339 | 158 | 34,813,120.4 | 485 | 1500 | 9 |

* Time in seconds.

**Table 5.** PMS for the proposed model.

| Schedule | Total Production | | The Lowest Gap | |
|---|---|---|---|---|
| | Electricity (MW) | Water (MIGD) | Electricity (MW) | Water (MIGD) |
| The proposed model | 17,640,000 | 36,128.4 | 42,484.4 | 3.96 |

Through these experiments, it can be concluded that water production and electricity will be affected in the event of increased demand in the future by more than 20% for water and 40% for electricity, without increasing the production capacity, by adding additional equipment, as well as replacing older pieces of equipment with new ones, since they require a longer maintenance period.

## 4. Conclusions

In this paper, a new nonlinear objective mixed-integer optimization model to generate an optimal PMS for power plant equipment consisting of boilers, distillers, and turbines, is presented. The aim of the study is to present a solution model that focuses on increasing the amount of production for water and electricity in addition to addressing a major issue, which is to increase the gap between demand and production during the maintenance weeks for the planning horizon. The data was taken from the Al-Zour station for the Ministry of Electricity and Water for the State of Kuwait, where the schedule for one year consists of 52 weeks. It is clear from the comparison between the proposed solution and the Ministry's schedule that there is a large difference in production for electricity in addition to the gap between production and demand, which was greater for both distillers and turbines. Through the sensitivity test—which increased the volume of water and electricity demand in addition to an increase in the maintenance duration of some older equipment—a risk was observed in the volume of production, which was reflected in the gap between production and demand. Therefore, it was proposed to replace older equipment, both for distillers and turbines, as well as to purchase additional equipment in anticipation of a future increase in demand for both water and electricity without risking a shortfall in meeting demand due to the failure of some equipment.

The extension of this research is the addition of the probability distributions of some of the deterministic parameters of the proposed model. This includes, for example, the probability of a set of equipment malfunctions and the cost of this outage or breakdown, as well as changing and increasing demands in the future.

Additionally, future work will address this problem using a metaheuristic method (GA) as a solution approach and will compare the results with the current study in terms of solution quality and computing time. Heuristic algorithms and the metaheuristic method are widely applied in various fields, such as medicine [47,48], transportation [49–52], online learning [53,54], data classification [55,56], etc., where they generate optimal or near optimal solutions in a short time.

**Author Contributions:** Data curation, Y.A.; formal analysis, K.A.; methodology, K.A.; writing—original draft, K.A.; writing—review and editing, Y.A. and M.F.A. All authors have read and agreed to the published version of the manuscript.

**Funding:** This research received no external funding.

**Data Availability Statement:** Not applicable.

**Acknowledgments:** I would like to express my sincere thanks and gratitude to the Public Authority for Applied Education and Training for its full support of this research (Project No. TS-21-15), project title: Nonlinear Programming for solving Preventive Maintenance Scheduling problem for Cogeneration Plants with Production. In addition, special thanks to Faisal Al-Shammari and to the Ministry of Electricity and Water in the State of Kuwait for providing us with information about the Al-Zour station.

**Conflicts of Interest:** The authors declare no conflict of interest.

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
