# Peer review of "Nonlinear Integer Programming for Solving Preventive Maintenance Scheduling Problem for Cogeneration Plants with Production"

_sustainability, doi:10.3390/su15010239_

Round 1
Reviewer 1 Report
The authors must write all steps for the derivation of the mathematical formulas.
Author Response
Many thanks for contributing to the review of my article and for your valuable feedback.
All notes have been modified and are highlighted in yellow.
Reviewer 2 Report
please see the attachment

Author Response
Many thanks for contributing to the review of my article and for your valuable feedback.
- Line 23-24 (modified and more explanation)
- Lines 45 (done)
- Line 129 (you have mentioned a good comment, but I think it’s better to be in the bigging).
- Line 153 (merged)
- Line 200 (done)
- Line 311 (removed)
- Line 336 ( I haven’t made any edits for fear of confusing the reader)
- Line 342 (done)
- Line 349 (formatted)
- Line 506 (done)
- References have been increased from 30 to 45.
Reviewer 3 Report
It is a work within a very interesting theme and with great importance for the sustainability not only of Kuwait but of the world. For the specific area of energy production with low production cost, this work is presented as important, however, some corrections must be made:
- In line 23 replace "," by "." in excerpt "... additional conditions"," Also, a comparison ..."
- Between lines 69 - 70, the authors used "swam" and "swarm", what is correct?
- In line 145 replace "," by ".";
- Between lines 200 - 243 need several corrections because exist many typographic mistakes;
- In the pdf file that I received the equations between lines 244 - 272 are much danified and difficult my analyze;
- In Table 3 correct the word "Electicity";
- In line 433 the figure legends are "inverted";
- In my pdf file, in lines 443 - 446 exist several mistakes.
Author Response
Many thanks for contributing to the review of my article and for your valuable feedback.
All notes have been modified and are highlighted in yellow:
- Line 23 (done)
- Lines 69-70 (done)
- Line 145 (done)
- Lines 200-243 (done)
- Lines 244-272 (done)
- Table 3 (done)
- Line 433 figure legends (done)
- Lines 443-446 (done)
Round 2
Reviewer 1 Report
No comments.
Author Response
Thanks for contribution.
Reviewer 2 Report
Ok
Author Response
Thanks for contribution.